# The Occurrence and Dietary Exposure Assessment of Mycotoxins, Biogenic Amines, and Heavy Metals in Mould-Ripened Blue Cheeses

**DOI:** 10.3390/foods9010093

**Published:** 2020-01-16

**Authors:** Ingars Reinholds, Janis Rusko, Iveta Pugajeva, Zane Berzina, Martins Jansons, Olga Kirilina-Gutmane, Kristina Tihomirova, Vadims Bartkevics

**Affiliations:** 1Institute of Food Safety, Animal Health and Environment “BIOR”, Lejupes iela 3, LV-1076 Riga, Latvia; janis.rusko@bior.lv (J.R.); iveta.pugajeva@bior.lv (I.P.); Zane.Berzina@bior.lv (Z.B.); Martins.Jansons@bior.lv (M.J.); Olga.Kirilina@bior.lv (O.K.-G.); Kristina.Tihomirova@bior.lv (K.T.); Vadims.Bartkevics@bior.lv (V.B.); 2Faculty of Chemistry, University of Latvia, Jelgavas iela 1, LV-1004 Riga, Latvia; 3Water Research Laboratory, Research Centre for Civil Engineering, Faculty of Civil Engineering, Riga Technical University, Kipsalas iela 6a-263, LV-1048 Riga, Latvia

**Keywords:** blue cheese, mycotoxins, biogenic amines, heavy metals, HPLC-MS/MS, HPLC-PAD, ICP-MS, dietary exposure, deterministic modelling, hazard index

## Abstract

The occurrence and dietary exposure assessment of 16 mycotoxins, 6 biogenic amines (BAs), and 13 metallic elements in blue-veined cheeses (*n* = 46) is reported. Co-occurrence of mycophenolic acid (≤599 µg·kg^−1^) with roquefortine C (≤5454 µg·kg^−1^) was observed in 63% of the tested cheeses, while BAs were frequently present at concentrations between 0.2 and 717 mg kg^−1^. The concentrations of heavy metals in cheeses were very low. Chronic/acute exposure assessment based on consumption data from different European populations indicated that the levels of mycotoxins and heavy metals are safe to consumers, whereas, rather high hazard indexes (HI up to 0.77) were determined for BAs according to the worst-case scenario based on high consumption and 95th percentile occurrence. A more detailed acute dietary intake study indicated that histamine and tyramine were predominant among these BAs, reaching 27 and 41% of the acute oral intake reference doses.

## 1. Introduction

Roquefort, Blue Stilton, Danablu, and Gorgonzola are highly appreciated blue cheese varieties, which have been traditionally produced in certain regions of France, England, Denmark, and Italy. The interest by European consumers in new flavours and local products has created incentives for the production of blue cheese products in Poland, Germany, Spain, and other countries. The presence of different fungi and biochemical transformations of milk proteins during the fermentation and ripening of cheese may promote the formation of mycotoxins and biogenic amines (BAs) [1,2].

From the viewpoint of safety, aflatoxin M_1_ (AFM_1_), the hydrolysed metabolite of aflatoxin B_1_ (AFB_1_), is the only regulated mycotoxin with an established maximum concentration of 0.05 µg kg^−1^ in milk used within the European markets [3]. Roquefortine C (ROQ C), mycophenolic acid (MPA), penicillic acid (PA), PR toxin, andrastatin A, and penitrem A (PNA) are major mycotoxins produced in blue cheeses mainly by strains of *Penicillium roqueforti* [4,5]. MPA and ROQ C have been commonly determined in blue cheeses by high-performance liquid chromatography with tandem mass spectrometry (HPLC-MS/MS) [6,7]. Ochratoxin A (OTA) and PNA have also been found in blue cheeses at 1–4 µg kg^−1^ levels [8,9].

BAs that are typically associated with improperly refrigerated or spoiled fish and meat may be found also in mould-ripened cheeses [10].

In Europe, permitted levels for histamine (His) in unprocessed fish have been established at 100–200 mg kg^−1^ and up to 400 mg kg^−1^ in fish sauce, while there are not regulations set for dairy products [11]. HPLC with pre-column derivatisation and UV/VIS or photo-diode array detection (PAD) have served as the main methods for the analysis of BAs in mould cheeses [12,13,14]. A recent study, based on HPLC-MS/MS with cation exchange column, has shown advantages for the analysis of ten BAs, while avoiding the derivatisation step [15].

Environmental and technological contamination by heavy metals capable of binding to milk proteins, can also affect the quality and safety of dairy products [16]. Heavy metal contamination can arise from air and water pollution in the regions of cattle farms, through cattle feed, or from technological stages in the dairy and cheese processing plants [17]. Microwave digestion combined with sensitive atomic absorption spectrometry (AAS), inductively coupled plasma with optical emission spectrometry or mass spectrometry (ICP-OES, ICP-MS) have been used for the analysis of heavy metals. Whereas, little information is available about such analyses of blue mould cheeses [18].

The aim of the current study was to evaluate multi-occurrence of mycotoxins, BAs, and heavy metals (both micronutrients and such toxic elements as Cd, As, Pb) in commercial blue-mould cheeses from the European Union (EU) Customs Union markets, and to integrate the available dietary exposure data into the risk assessment. Acute and chronic dietary exposure studies were taken into account according to three different scenarios based on the mean consumption/moderate occurrence values, high consumption/moderate occurrence, and the worst-case scenario of high consumption/95th percentile (P95) occurrence. The exposure analysis was performed separately for various populations across Europe, based on the data and guidelines available from the Comprehensive Food Consumption Database (hereafter referred to as the EFSA database), provided by the European Food Safety Authority [19].

## 2. Materials and Methods

### 2.1. Materials and Chemicals, Reagents, and Standards

Mycotoxin standards were all of at least 95% purity. OTA, AFB_1_, AFM_1_, and citrinin (CIT) were obtained from Romer Labs (Tulln, Austria). ROQ C, PA, and MPA standards were supplied by Santa Cruz Biotechnology (Dallas, TX, USA). Ochratoxin B (OTB) and PNA were purchased from Fermentek (Jerusalem, Israel). Beauvericin (BEA), citreoviridin (CVD), four enniatins (ENN A, ENN A_1_, ENN B, ENN B_1_), and stachybotrylactam (SBL) were obtained from Cayman Chemical Company (Ann Arbor, MI, USA). Analytical standards of six BAs (>97% assay), namely, cadaverine (Cad), histamine (His), 2-phenylethylamine (Phe), putrescine (Put), tryptamine (Try), and tyramine (Tyr) were obtained from Sigma-Aldrich (Steinheim, Germany) that also provided hexane (pesticide grade) and acetone (HPLC grade). Multielement standard solution V for ICP in 10% nitric acid (TraceCERT^®^), HPLC grade acetonitrile and methanol (>99% assay), ACS grade formic acid (≥96.0% assay), hydrochloric acid (37%), and dansyl chloride (BioReagent grade, ≥99.0% assay) were also purchased from Sigma-Aldrich. Nitric acid (≥69.0% assay) for trace analysis was obtained from Fluka (Buchs, Switzerland). Hydrogen peroxide (30%) of extra purity was purchased from Merck (Darmstadt, Germany). Sodium bicarbonate was obtained from Scharlab (Barcelona, Spain). Ultrapure water (18.2 MΩ × cm) was generated by a Milli-Q system (Millipore, Billerica, MA, USA). QuEChERS buffer–salt extraction kits consisting of magnesium sulphate (4 g), sodium chloride (1 g), trisodium citrate dihydrate (1 g), and disodium hydrogen citrate sesquihydrate (0.5 g) per portion, dispersive solid phase extraction (DSPE) tubes (roQ™, 15 mL) containing magnesium sulphate (900 mg), primary secondary amine (150 mg) and C18E silica (150 mg) per kit, and Phenex™ RC syringe filters (0.45 µm) were purchased from Phenomenex (Torrance, CA, USA).

### 2.2. Cheese Samples

A total of forty-six blue cheese samples were purchased from supermarkets and gourmet stores in Riga, Latvia. Three of the samples were from Latvian dairies, while the rest were from Denmark (*n* = 7), England (*n* = 3), France (*n* = 9), Germany (*n* = 2), Italy (*n* = 11), Lithuania (*n* = 2), Poland (*n* = 7), and Spain (*n* = 2). The sample weight ranged between 40 and 200 g. Brine-matured fresh cheese “Grikios” (200 g, 45% fat content per dry matter, EUROSER Dairy Group, Warszawa, Poland) was used as the blank matrix in method validation for the analysis of mycotoxins and BAs. The samples were immediately transported to the laboratory, homogenised, placed in bags (Whirl-Pak^®^, Nasco, Kellogg, ID, USA), and stored at −20 °C until analysis.

### 2.3. Mycotoxin Analysis

#### 2.3.1. Standard Solutions and Sample Preparation

Stock solutions of mycotoxins were prepared by dissolving solid standards (approximately, 10 mg of each) in the appropriate solvents (10 mL). The working standard solutions of each mycotoxin were prepared at different levels (the concentration ranges are summarised in Appendix A) and stored at −20 °C for no longer than 1 month. Once thawed, the pre-weighed (30 g) samples were degreased for 10 min by hexane (15 mL) on a mechanical shaker (Multi RS–60, BioSan, Riga, Latvia), and centrifuged (3063× *g*, 10 min). The hexane (upper) phase was discarded and the lower phase extracts were used. Defatted samples (5.00 ± 0.01 g) were accurately weighed in 50 mL PP tubes. The quality control (blank) samples were supplemented with mycotoxin standard solutions at the appropriate spiking levels. Then water (10 mL), acetonitrile (10 mL), and formic acid (20 µL) were gradually added to the tubes and extraction was started by mixing for 5 min on mechanical shaker. One portion of the QuEChERS buffer salt kit was added to each of the tubes and the extraction was continued for additional 10 min. The obtained mixtures were centrifuged (1313× *g*, 5 min) and the supernatants were transferred to 15 mL centrifuge tubes and stored for 15 min at −80 °C in a Heto PowerDry^®^ freeze dryer (Thermo Fisher Scientific, Waltham, MA, USA). After removal, the extracts were immediately centrifuged (2626× *g*, 5 min) at 10 °C. For each sample, replicate volumes (500 µL) were transferred to 10 mL glass tubes, whereas, the remaining extracts (5 mL) were transferred to QuEChERS dSPE centrifuge tubes for a clean-up. The tubes were shaken for 5 min and centrifuged (2626× *g*, 5 min) at room temperature to obtain purified extracts. The initial fractions (500 µL) and the purified extracts (3.5 mL) were pooled and evaporated to dryness at 50 °C under a gentle nitrogen stream. The dry residues were reconstructed in 150 µL of injection solution and transferred into the autosampler for analysis. In the case of mouldy samples, they were filtered through centrifuge filters (390× *g*, 10 min) before the analysis.

#### 2.3.2. Chromatographic Method for Mycotoxin Analysis

The analysis was performed on an UltiMate™ 3000 (Thermo Fisher Scientific, USA) HPLC coupled with a Thermo Scientific TSQ Quantiva MS/MS detector. The separation was performed on a Phenomenex Kinetex™ C18 reversed-phase analytical column (50 × 3.0 mm, 1.7 µm). A ternary gradient elution was carried out using 0.1% formic acid in water (eluent A), 0.1% formic acid in methanol (eluent B), and 0.1% formic acid in acetonitrile (eluent C) according to the following gradient program: 0–1.5 min: 0% B and 30% C; 2.0–2.7 min: 15% B and 35% C; 5.5–6.5 min: 40% B and 58% C; 8.0 min: 5% B, 93% C; 8.5–9.5 min: 0% B and 10% C; 10.0 min: 0% B and 30% C. The flow rate was 0.35 mL/min. The autosampler was maintained at 4 °C and the column temperature was 40 °C. The sample injection volume was 15 µL. Ion monitoring was conducted in both positive and negative ion modes and the mass analysis was performed in selected reaction monitoring (SRM) mode. The following instrumental settings were used: spray voltage 3.5 kV (positive ion mode), 2.5 kV (negative ion mode), vaporiser temperature 350 °C, ion transfer temperature 300 °C, sheath gas 55 arbitrary units (arb), auxiliary gas 25 arb, and sweep gas 5 arb. Data processing was performed with Xcalibur™ software (Thermo Fisher Scientific). Instrumental parameters are shown in Table 1.

#### 2.3.3. Validation of the Method

Validation was performed according the guidelines of Commission Regulation No 519/2014 [20] and included a check for linearity, limit of detection (LOD), limit of quantification (LOQ), recovery (R, %), repeatability (RSD, %), and method uncertainty based on blank matrix (fresh cheese “Grikios”) sample evaluation spiked with mycotoxin standards. To evaluate the linearity, six-point calibration curves were constructed using blanks spiked with mycotoxin standard mixtures. The least squares regression method was used for slope construction and calculation of the determination coefficients (R^2^) of the calibration curves, which were evaluated to a fit of at least 0.99. For quality control purposes, the blank samples were spiked with mycotoxin standard solvents at the following concentration levels: at 0.02, 0.06, and 0.1 μg kg^−1^ for AFM_1_; at 1, 10, and 18 μg kg^−1^ for ENNs, at 1, 10, and 20 μg kg^−1^ for AFB_1_, OTA, OTB, STB, PNA, and BEA; at 10, 30, and 70 μg kg^−1^ for CIT; at 10, 50, and 100 μg kg^−1^ for PA; and at 5, 50, and 250 μg kg^−1^ for ROQ C, MPA, and CVD. The precision, repeatability, and recovery studies within intra-day tests (two days within a one-week period) were performed by analysing six replicates at each of three determined spiking levels. The mean values of each replicate were determined at three concentration levels and compared to the acceptance criteria specified by the EC regulation No 519/2014 [20]. The signal-to-noise (S/N) approach was used to estimate the limit of detection (LOD) and the limit of quantification (LOQ). The chromatographic noise and analytical response were estimated using the chromatograms of spiked samples. The LODs and LOQs were defined based on the signal (S) to noise (N) ratios: S/N > 3 and S/N > 10, respectively. The measurement uncertainties were calculated according to Eurachem/CITAC Guide CG 4 [21]:(1)$$U=k\times u_{c}=k\times \sqrt{u^{\prime }{\left(wR\right)}^{2}+u^{\prime }{\left(bias\right)}^{2}}$$
where $u_{c}$ is the combined standard uncertainty, $u^{\prime }$*(wR)* is the within-laboratory reproducibility, $u^{\prime }$*(bias)* is the uncertainty component arising from method and laboratory bias, U is the expanded measurement uncertainty, and the coverage factor *k* = 2 at 95% confidence level [21].

### 2.4. Determination of Biogenic Amines

#### 2.4.1. Standard Solutions and Sample Preparation

Individual standard solutions of biogenic amines (1 mg mL^−1^) were prepared by adding accurately weighed amount of every BA standard (100 mg) to a 100 mL volumetric flask and diluting to the mark with a 0.1 M hydrochloric acid.

The final stock solution was prepared by mixing individual solutions and was used for spiking of blank samples for the method development and quality control experiments.

Sample preparation was conducted in darkness due to concerns about the stability of dansyl chloride used as derivatisation agent. A 5 mg mL^−1^ solution of dansyl chloride in acetone was prepared by adding accurately weighed 50 mg of dansyl chloride to a 10 mL volumetric flask and diluting to the mark with solvent. Homogenised cheese samples (5.00 ± 0.01 g) were weighed into 50 mL polypropylene tubes and pre-treated with 20 mL of 0.1 M hydrochloric acid. For quality studies, the acidified blank samples were spiked with standard solutions to 1, 20, and 58 mg kg^−1^ concentrations. For linearity testing, six calibration samples were prepared by adding the standard solutions to acidified blank matrix until BA levels of 1.0, 5.0, 10, 25, 50, and 75 mg kg^−1^ were reached. The mixtures were agitated for 50 min on a mechanical shaker and centrifuged at 2626× *g* for 10 min at room temperature. A 10 mL volume of the extract was transferred to a new tube and supplemented with 0.1 M hydrochloric acid (10 mL), followed by the addition of saturated sodium bicarbonate solution (10 mL). The tubes were capped and vigorously shaken, the extracts were then filtered through Phenomenex RC syringe filters (0.45 µm). Aliquots of 500 µL were transferred to 15 mL PP tubes, followed by addition of dansyl chloride solution in acetone (500 µL). Capped tubes containing the samples and derivatisation agent were vigorously shaken. The tubes were then opened and placed into a heating block and heated for 1 h at 40 °C, while agitating every 15 min. After cooling to room temperature, the mixtures were diluted with diethyl ether (2 mL) and the tubes were placed on mechanical shaker for 10 min extraction and then centrifuged for 10 min (2626× *g*) at 10 °C.

Extract replicates (1 mL) from the diethyl ether (upper) layer were transferred to new 15 mL PP tubes, placed in water bath and evaporated at 40 °C to dryness under a gentle stream of nitrogen. For each sample, the dry residue was reconstituted in 500 µL of acetonitrile/water (30:70, *v/v*) solution. The resulting extracts were transferred to autosampler glass vials for further analysis using HPLC-PAD method.

#### 2.4.2. Method for the Determination of Biogenic Amines

An UltiMate™ 3000 (Thermo Fisher Scientific, Waltham, MA, USA) high-performance liquid chromatograph was combined with a Thermo Scientific PAD detector. Separation was performed on a Phenomenex Luna Omega C18 reversed-phase analytical column (100 × 2.1 mm, 1.6 µm). The gradient elution was carried out using a mixture of water (eluent A) and acetonitrile (eluent B), according to the following program: 0 min: 30% B; 21.0 min: 62.7% B; 21.3 min: 97.5% B; 28.8 min: 97.5% B; 29.0 min: 30% B. The wavelength range of 190–800 nm was scanned and the chromatograms of BAs were recorded at 340 nm, which was identified as the optimal wavelength for the analysis.

#### 2.4.3. Validation of the Method

Fresh cheese “Grikios” was also used as the best suitable blank matrix within BA method validation, as it was free of biogenic amine contamination while having similar physicochemical characteristics to tested cheese varieties. All of the necessary validation parameters were determined on the basis of calibration studies and inter-day studies of blank samples spiked to three concentration levels (1, 20, and 58 mg kg^−1^) within a two-day period (5 replicates per sample were evaluated). The same spiked samples were used within the recovery studies and for the determination of inter-day RSDs. Linearity was determined from six-point calibration curves constructed by plotting the chromatographic peak areas of BAs against the respective spiked concentrations (1 to 75 mg kg^−1^) of standards. The equation of determination was calculated for each of the calibration curves and the values of *r*^2^ > 0.99 were assumed as acceptable for the linearity.

In the case when the concentrations exceeded the linear calibration range, a dilution procedure was used to ensure correct determination of BA concentrations.

For the evaluation of method sensitivity, additional blanks were spiked with standard at 1 mg kg^−1^ level. The standard deviations (SDs) of four blank sample replicates spiked to 1 mg kg^−1^ were calculated and used for the determination of LOD and LOQ values of the provided method: SD × 5 and SD × 10 were used for the determination of LOD and LOQ, respectively. The expanded method uncertainty for each BA was calculated from the standard uncertainties *u_c_* obtained from the charts of quality control results (data of more than 28 replicate studies that were multiplied with the coverage factor (*k* = 2)).

### 2.5. Determination of Heavy Metals

#### 2.5.1. Sample Preparation

A slight modification of previously reported method [22] was applied for the determination of trace elements and toxic metals in mould-ripened cheeses. After thawing, a 0.5 g portion of each sample was weighed in a PTFE digestion vessel by using analytic balance (precision ± 0.0001 g). Three replicates were tested for each of the cheese samples. Deionised water (2 mL), 65% nitric acid (5 mL), and 30% hydrogen peroxide (3 mL) were then added to each sample. The vessels were kept at room temperature for at least 20 min to complete the reaction. Then the vessels were capped and transferred to a Mars 6 microwave oven (CEM corporation, Matthews, NC, USA) for digestion. The temperature was raised to 150 °C within 15 min and was held for 15 min at 150 °C, then raised to 180 °C within 10 min and held at that temperature for 20 min. After cooling, the vessels were opened to release the pressure of evolved gaseous products.

The samples were filtered through a filter with 22 μm pore size (Whatman, Little Chalfont, UK), quantitatively transferred to volumetric flasks and diluted to 25 mL volume with deionised water. Blanks were prepared by the same procedure that was used for the analysed samples. The standard solutions used in the construction of calibration curves for quantification studies were prepared at the concentration levels of 1, 2, 5, 10, 50, 100, 500, and 1000 μg L^−1^ using a stock solution containing 10 mg L^−1^ of each target element per 100 mL.

#### 2.5.2. Determination of Heavy Metals

An Agilent 7700× ICP-MS instrument with Mass Hunter Workstation software for ICP-MS, version B.01.03 (Tokyo, Japan) was used for the analysis of thirteen elements: Aluminium (Al); microminerals iron (Fe), manganese (Mn), cobalt (Co), nickel (Ni), copper (Cu), zinc (Zn), selenium, and molybdenum (Mo); and heavy metals tin (Sn), arsenic (As), cadmium (Cd), and lead (Pb). The optimised operating conditions are summarised in Table 2. The concentrations of the metallic elements were calculated from the obtained external calibration lines, with each line constructed from eight calibration points at 1–1000 μg L^−1^ concentration levels and an additional blank sample measured under the same conditions and adjusted by least-squares regression analysis. The final results are the average of three replicates.

#### 2.5.3. Validation of the Method

The nine-point calibration curves (eight standards and one blank) were constructed for each of the metal ions and the correlation coefficients were >0.999 before starting the sample analysis. The limits of detection (LODs) were individually calculated as three-fold standard deviation of blanks and the limit of quantification (LOQ)—as ten-fold standard deviation of blanks for each set of measurements. Daily analyses of quality control materials (with the concentrations of 2.00 μg/L and 50 μg/L) were used for monitoring the repeatability and accuracy, as well as determining the method uncertainties according to the Equation (1). The calculation of the individual expanded method uncertainties for the individual chemical elements was provided using data of interlaboratory test results based on control reference materials.

### 2.6. Dietary Exposure Assessment

#### 2.6.1. Food Consumption Data

Exposure assessment was performed using food consumption data from the EFSA database, built on the basis of a collaborative research agreement between the EFSA and the European Union member states to pool national-level data collected through nationally representative surveys in each country [19,23]. The use of the EFSA database was found suitable for the current study, in line with the recommendations for the use of these statistics to assess dietary exposures to hazardous substances [19]. The data retrieved from the surveys were filtered to fit the necessary significance of P95 values, with more than 60 observations required. The available amount of chronic consumption data was more limited, so the requirement was set to more than 30 observations per consumption entry.

The EFSA database was accessed at the public level to obtain the overall statistics for the consumption of blue-cheese varieties across Europe under the FoodEx2 “Exposure hierarchy” level 5, including the following cheese types: “Firm-ripened blue mould-veined cheese” and “Soft-ripened cheese veined with blue mould (e.g., Blue Bavarian and Blue de Graven varieties)”. All of the blue cheese subgroups were included.

Data were retrieved from ten surveys (Appendix A). Short-term consumption data for various types of blue cheeses were obtained from eight surveys, while six surveys provided data about the long-term consumption of such products. The particular design of each consumption survey and the obtained consumption data (in g × kg^−1^ bw × day^−1^) are available in the Appendix A. The consumption habits were recorded with 3- and 7-day food records, as well as 24-h and 48-h recalls. Most of the surveys were conducted between years 2000 and 2009, while two surveys were from years 2012 and 2014. Data recorded from the consumption days only was used for the assessment of short-term exposure. For the assessment of long-term exposure, the “consumers only” dietary intake of blue cheese was used.

#### 2.6.2. Exposure Assessment and Risk Characterisation

Dietary exposure to heavy metals, BAs, and mycotoxins was assessed according to a multistep procedure. Since the data collected during different dietary surveys could not be merged together, the exposure was assessed at national and age-specific levels [19].

A deterministic approach was used to calculate point estimates of dietary exposures [19]. Acute exposure was calculated for biogenic amines and selected mycotoxins (ROQ C, MPA, AFM1, and ENN B), while chronic exposure estimates were calculated for heavy metals. A conservative approach was applied to the occurrence data and all cases of non-detection were replaced with the LOQ values for the particular analytes, in order to obtain the upper-bound values according to EFSA recommendations for dietary exposure evaluation methodology [24].

The consumption data including the mean and P95 values were retrieved from the EFSA database, the Food Consumption Data section. The mean and upper-bound values of chronic exposure to heavy metals were obtained by combining the mean and P95 consumption (high consumption), respectively, with the mean and upper-bound concentrations of heavy metals.

Three types of scenarios were investigated for the purpose of acute exposure assessment. According to the first two scenarios, the mean and upper-bound values of acute exposure to BAs and mycotoxins were obtained by combining the mean and P95 consumption with the mean and upper-bound concentrations of the respective contaminants [19]. Additionally, the third, worst-case scenario was based on the P95 consumption data in combination with the P95 consumption levels of the respective BAs and mycotoxins.

The resulting levels of exposure to heavy metals and BAs were assessed by comparing with the oral reference doses for toxic effects (RfD and aRfD values for acute effects in the case of BAs). Following a generally recognised approach for risk assessment, the risk characterisation step was executed through the calculation of hazard quotients (HQ) and hazard indices (HI) as ratios between the actual exposure and the established RfD and aRfD values. The HQ values were calculated separately for individual contaminants. The HI values were used to separately characterise the groups of heavy metals and BAs by summing up the individual HQ values (Equation (2)):(2)$$HI=\overset{n}{\underset{k=1}{\sum }}HQ_{k}=\overset{n}{\underset{k=1}{\sum }}\frac{C_{k}\times Consumption}{aRfD_{k}}$$
where $C_{k}$ is the determined concentration of contaminant k, Consumption is the dietary intake by the surveyed population, *aRfD* is the reference dose of acute toxic effect threshold for compound k. As generally accepted, HI < 1 indicates a tolerable (safe) exposure level and HI > 1 indicates unacceptable exposure level according to the recommendations of the United States Environmental Protection Agency [25].

Additionally, for the characterisation of risks due to biogenic amines, we assessed the percent contribution of acute dietary exposure to the aRfDs for six individual BAs, in order to better identify the major contributors to possible toxicological effects.

The deterministic approach used in this study could result in the overestimation of exposure, but this tendency can be considered to be acceptable for a screening assessment [26].

## 3. Results and Discussions

### 3.1. Method Validation

All of the methods provided sufficient linearity, with the determination coefficients (R^2^) higher than 0.99 for all of the tested analytes.

The mycotoxin method showed satisfactory recoveries (R = 91–117%) and intra-day repeatability (RSDs < 20%, with the exception of 29% for BEA) and acceptable measurement uncertainty for most of the target analytes (6–20%), whereas, higher uncertainty values (>20%) were determined for OTA, CVD, ENNs, and BEA (Appendix A). These data were obtained from the validation using brine-matured fresh cheese as the blank matrix. Compared to other reported methods, the sensitivity of the developed method was satisfactory—the LODs for AFM_1_, MPA, and ROQ C (0.004, 0.12, and 0.47 µg kg^−1^) were at least 5 times lower compared to other methods, for example, as described by [7], whereas the detection limits for these mycotoxins were 0.02, 3.00, and 4.00 µg kg^−1^.

The brine-matured fresh salad cheese was successfully used as the blank matrix also for quality control of HPLC-PAD method, as this type of cheese has been reported to contain notably lower BA levels compared to other cheese varieties [10,27]. Satisfactory recovery (83–117%), precision, and uncertainty (RSD < 16%) of the HPLC-PAD method were determined from the performance assessment studies involving the analysis of samples fortified with a standard additive at three concentration levels, five replicates at each level, with the analyses performed over two days (Appendix A). The method uncertainties were determined from the control chart data and were in the range of 26–32% (Appendix A).

The selectivity of the HPLC-PAD method was tested by evaluating the effects of analyte signals and interfering signals generated by the matrix. As shown in Figure 1, no peaks interfered with the analyte signals. Although the retention times of Cad and His were close, selectivity was sufficient, as determined at the highest calibration point (75 mg kg^−1^).

The stability of BAs was evaluated within a single sequence by measuring the change of analyte concentrations over one sequence (~20 injections). Samples were stored in the autosampler at 15 °C in the dark for the analysis (Appendix A). While the concentrations of Phe, Try, and Tyr showed slight decrease during the sequence and the concentrations of other BAs increased by a small amount, the fluctuations were only in the range of 0.2–2.6%, indicating that the analytes remained stable during the sequence, thus allowing their proper determination by the developed method. In addition, the content of BAs in some cases exceeded the maximal value (75 mg kg^−1^) of the linearity range. Therefore, if the concentration exceeded the calibration range, the samples were diluted and the actual levels were recalculated accordingly. This procedure also allowed to avoid specific non-linear calibration issues that may occur at the high concentrations used in BA testing involving derivatisation with dansyl chloride [28].

The performance parameters of ICP-MS method for heavy metals analysis are summarised in Appendix A, which indicate sufficient recovery (R < 120%), repeatability (RSDs < 20%) according to the requirements of EU legislation [29]. While the sensitivity was lower than achieved by [18], the LOD values were sufficient, compared to the EU regulations for the determination of toxic elements.

### 3.2. The Occurrence of Mycotoxins in the Tested Blue Cheeses

The developed HPLC-MS/MS method enabled simultaneous analysis of mycotoxins produced by starter cultures (ROQ C, MPA) and specific strains that can indicate spoilage within one sample injection. As expected, all of the samples contained detectable concentrations of ROQ C ranging between 3.2 and 5.454 µg kg^−1^ (Table 3).

The lowest ROQ C content was determined in Morbier AOC French cheese variety, which was analysed for comparison, reflecting the fact that this semi-hard cheese does not belong to the blue-veined variety. As noted in Table 3, the concentrations of ROQ C in cheeses from Poland ranged from very low to medium high levels. The lowest and highest concentrations (5.5 and 790 µg kg^−1^) were associated with specific cheese varieties, namely, brie (white-mould cheese) and blue-veined cheese. The mean ROQ C levels determined in other blue cheese samples from Poland ranged between 250 and 590 µg kg^−1^. The highest content of ROQ C was found in Blue Stilton varieties, with the mean concentration of 3675 µg kg^−1^.

Latvian Bio-Blue, three Italian, and one German cheese, as well as the majority of Danish blue cheeses also contained rather high contents of ROQ C. The differences of ROQ C content could be attributed to different concentrations of fungal starter cultures added to cheeses during their production, rather than differences in ripening [7]. An earlier study, indicated 1.5–12 times higher concentrations of ROQ C (0.8 to 12 mg kg^−1^) found in blue cheeses from the Finnish market as compared to that of the current study [6]. However, similarly to our test results, French cheeses in that study showed a widest variation, whereas only one of the total of twenty-one samples from the Finnish market was positive for MPA, with the concentration of 300 µg kg^−1^ [7].

The current study of forty-six cheese varieties marketed in Latvia showed co-occurrence of MPA with ROQ C in 63% of the samples (*n* = 29), with the concentrations of MPA ranging between 5.8 and 599 µg kg^−1^ (Table 3). The highest concentrations of MPA (>190 µg kg^−1^) were found in the Danish blue (*n* =3) and Roquefort type (*n* = 3) cheeses, whereas eighteen of the samples, mainly originating from Poland, Italy, Germany, Latvia, and in three cases from Denmark contained relatively low amounts of MPA (<50 µg kg^−1^). The concentrations of MPA in two of the Danish cheese samples analysed during our study were in agreement with those reported by other authors, reaching 500 µg kg^−1^ of MPA in these cheese varieties according to HPLC-UV/Vis analysis [30]. Our results were comparable with the mean mycotoxin levels reported for blue cheese varieties by [7], who tested more than eighty commercial cheese samples and evaluated the evolution of ROQ C and MPA concentrations in the samples during different ripening stages. The determined mean average levels of ROQ C and MPA from those studies were in the range of 11–14,125 µg kg^−1^ and 15–6190 µg kg^−1^, respectively. In that study, the cheese samples were categorised according to the origin of milk. While higher levels of MPA were present in goat milk and the lowest levels were observed in cheeses produced from blended milk (there was little change for ROQ C), no conclusions about correlation could be made due to the scarcity of data about different cheese categories. That was also noted during our study, as the cow milk samples showed rather wide distribution of mycotoxin levels compared to the mixed or other milk types. Furthermore, according to the study by [7], MPA concentrations have a tendency to increase with ripening time.

Most of the recently reported methods have been focussed on a particular mycotoxin or group of mycotoxins such as OTA [8,31], penitrems A to F [9], whereas AFM_1_ was mainly studied together with ROQ C and MPA [6,7,32]. The transformation of labile mycotoxins (PA, PR toxin) into metabolites during storage has also been characterised, whereas an absence of those mycotoxins in manufactured blue cheeses was reported [33]. Cacmacki et al. (2015) reported rather high concentrations of PA (0.2–43.6 mg kg^−1^), as well as similar levels of ROQ C (0.4–47.0 mg kg^−1^) and MPA (0.1–23.1 mg kg^−1^) in Turkish traditional cheese, which was ripened for a three-month period in the presence of *P. roqueforti* fungal cultures [34].

The current study did not indicate elevated concentrations of PA or other mycotoxins that are especially associated with spoilage (e.g., SBL, OTs, CVD, and others). AFM_1_ was also below the detection level in most of the tested samples, indicating a good quality of milk used for the cheese production, except for three out of five Italian Gorgonzola samples that contained detectable concentrations of AFM_1_, from 0.02 µg kg^−1^ up to the maximum permitted level of this toxin in milk (0.05 µg kg^−1^). For comparison, the study [7] and other earlier reports did not report detectable levels of AFM_1_.

An interesting finding within the current study was the detection of one emerging *Fusarium* mycotoxin (ENN B), which was present at low concentrations (1.0–1.7 µg kg^−1^) in four different cheese samples made of cow milk—two French cheeses, one sample from Poland, and processed blue veined cheese from Lithuania. No other ENNs or BEA were found. This is the first study indicating traces of this emerging mycotoxin in blue mould cheese made of cow milk. A recent paper by Polish researchers based on HPLC-MS/MS method revealed low levels of ENN B (0.0055–0.0121 µg kg^−1^) in eighteen out of the total of twenty tested samples of sheep milk [35]. Taking into account the recent report by Tolosa et al. (2019) of ENN B as the most common emerging mycotoxin in animal feed [36], it is possibly worth to continue studies of emerging *Fusarium* mycotoxins in food and feed, including foods of animal origin, such as dairy products.

### 3.3. The Occurrence of Biogenic Amines in Blue Cheeses

The results of BA determination are summarised in Table 4 and Figure 2 as the frequency of BA occurrence in the tested cheeses at different concentrations.

Concentrations below LODs were found in only one out of six tested Gorgonzola samples. Two samples (Italian Montagnolo and Lithuanian processed blue cheese) contained His at concentrations above the method LOQ (0.2 and 0.3 mg kg^−1^). However, these two samples also contained Tyr (440 and 50.3 mg kg^−1^). The other tested cheese samples were positive for one to six BAs, with the individual concentrations ranging from just above the reporting level (>1 mg kg^−1^) up to 719 mg kg^−1^, while the total content of BAs ranged between 5.5 and 824 mg kg^−1^. Almost quarter (*n* = 12) of the tested cheese samples contained under 10 mg kg^−1^ of BAs (Figure 2).

In other samples, one or two BAs exceeded that level. Five samples, including two Cambozola, Monatanaglo, Gorgonzola and Ranka Blue cheeses, and Latvian blue cheese were found with Tyr, Try, and His concentrations exceeding 150 mg kg^−1^ (Table 4). The rather wide differences of BA distribution among the tested cheese samples were influenced by the different geographical origins and types of processing (Figure 2). A risk evaluation of BAs in different food products was reported by EFSA in 2011, covering more than 600 blue cheese samples from different EU countries [27]. This EFSA report indicated that cadaverine (Cad), as BA of low oral toxicity, was the most prevailing contaminant of this type in 23% of the reported samples (mean levels: 83.1–121 mg kg^−1^). Within this study, the presence of His was detected in 15% of the samples at the concentration range of 21.8–63.8 mg kg^−1^, Tyr was detected in 17% of the cases (63.2–104 mg kg^−1^), and Put was also found in 17% of the cases (20.9–62.2 mg kg^−1^). Other, less toxic BAs (e.g., Try, Phe) have been detected at rather low concentrations, but occurred frequently [13,27]. According to deterministic risk characterisation based on the hazard level (NOAEL) of 50 mg set for His by EFSA and the 270 g per day upper limit of consumption, 185 mg kg^−1^ is the highest tolerated concentration of His in healthy adults not using antihistamine drugs [27]. Benkeroum (2016) discussed different approaches to risk evaluation, used in Australia and other countries, which all included levels ranging from 100 to 400 mg kg^−1^ as the maximum allowed levels according to EFSA recommendations for healthy adults not using antihistamine drugs [10]. However, these values should be re-evaluated, taking into account the different consumption data in Europe, and this task is part of the dietary assessment in the current study.

Phe was the most prevalent BA, detected in more than 84% of the tested forty-six cheese samples in the current study. Similarly, to other BAs, Phe is known to act as a neurotransmitter, which can initiate hypertension and headache, with a threshold toxic level of 30 mg in healthy adults not under medications [10]. The concentration of Phe ranged between 1.5 and 21.6 mg kg^−1^ in the tested blue mould cheeses, with an exception of 43.9 mg kg^−1^ determined in processed blue cheese from Lithuania. For comparison, according to the recent review [10], the concentrations of Phe ranged from below the limit of detection to 39.7 mg kg^−1^ found by a research study in 2003, whereas, more recent studies of cheese from sheep’s milk found Phe levels up to 69 mg kg^−1^ [14]. An earlier report by EFSA claimed the detection of Phe in 54% of the cases, which is less frequently than in the current study, while the concentrations for Phe in the current study were quite similar to those reported by an EFSA survey of blue veined cheese with the mean and the highest concentrations reaching 5.5 and 39.5 mg kg^−1^, respectively [27].

In the current study, His was the second most prevalent BA, which was found at concentrations between 0.2 and 186 mg kg^−1^ in 69.6% of the tested cheese samples. Mayer et al. (2018) recently reported the detection of His in 19 out of 31 samples (61% of samples) at the mean concentration of 36.6 mg kg^−1^, with the maximum level of 255.3 mg kg^−1^ [13]. This study determined that higher levels of His may be presented in acid-curd and hard cheeses, with the maximum concentration of 1159.7 mg kg^−1^.

Madejska et al. (2018) reported potential safety issues during the storage of white mould and blue-veined cheeses purchased in Poland [37]. In the case of white mould cheese, an increase of His content from 150 to 400 mg kg^−1^ was found after long term storage in a conventional refrigerator at +4 °C. The level of His in Gorgonzola Piccante reached 730 ± 20 mg kg^−1^ after storage for forty-two days at +22 °C. The authors of that study noted rather high levels of His (162.63 ± 23.57 mg kg^−1^) in raw Gorgonzola Piccante control samples, implying improper storage conditions during the processing or ripening of the tested cheese sample. These observations by Polish researchers were in agreement with the current research, where six different Gorgonzola samples were included. Four of the Gorgonzola varieties examined during our study contained His in the range from <LOD to 63.6 mg kg^−1^, but two samples of Gorgonzola Piccante showed elevated His contents: 140 and 186 mg kg^−1^.

Tyramine, an aromatic amine arising from the decarboxylation of tyrosine is commonly present in cheese samples at higher concentrations compared to other BAs. The typical concentrations in the range from 100 to 800 mg kg^−1^ [10] are acceptable for healthy adults. In our study, 58.7% of the samples (*n* = 27) were positive for Tyr at concentrations generally ranging from 1.1 to 110 mg kg^−1^ in 25 of the samples, whereas, two samples of Italian cheeses showed rather high concentrations of this BA: the previously reported Montagnolo cheese with Tyr content of >440 mg kg^−1^ and Cambozola with garlic containing >700 mg kg^−1^ of Tyr. These data are in agreement with the results reported by other researchers, which also noted high Tyr levels in Italian and other mould-ripened cheeses: 627 mg kg^−1^ [38], 308.7 mg kg^−1^ and 308.65–585.47 mg kg^−1^ in cheese made from sheep milk [14].

The levels of Cad, Put, and Try in the tested cheese samples were very similar, with 22 to 23% of samples containing detectable concentrations of these BAs, ranging between 1.2 and 213 mg kg^−1^. Compared to His and Tyr, these three BAs have low acute toxicity. The acceptable levels of Cad and Put in cheese are 540 mg kg^−1^ and 180 mg kg^−1^, respectively [10]. Our data indicated considerably lower levels of Put and Cad, with the maximum content of these BAs equal to 45.5 and 131 mg kg^−1^ in Italian and Polish cheese samples. A Gorgonzola cheese from Italy had a rather high (197 mg kg^−1^) content of Try in our study as well. For comparison, Mayer et al. (2018) found up to 110 mg kg^−1^ of Try only in 1 out of 31 tested commercial cheese samples, whereas Put and Cad where found in 29 and 16 out of the total of 31 blue cheeses, with the maximum levels of 527 and 830 mg kg^−1^, respectively [13].

### 3.4. The Occurrence of Chemical Elements, Microelements, and Toxic Metals

Only forty-four of the cheese samples were selected for heavy metal analysis, as two cheese samples from Poland were insufficient for these tests. The results are shown in Table 5. Among the determined heavy metals, cobalt (Co), nickel (Ni), copper (Cu), and arsenic (As) were below the levels of detection (Table 5). Christophoridis et al. (2019) also reported the absence of as even at trace levels (<1.1 µg kg^−1^ wet weight (ww)) in blue mould cheese marketed in Thessaloniki, Greece [18]. The absence of As in this case was reasonably expected, as its most commonl source is water from polluted geographic regions that is consumed by animals [39]. In fact, the study by Christophoridis et al. (2019) is the only study of heavy metals in blue mould cheeses. That study reported mean concentrations of Cu and Ni reaching 290 and 485 µg kg^−1^ ww in blue mould cheese, whereas Ni levels were associated with contamination through storage tankers and milking equipment within the primary processing stages.

Similarly, to lead (Pb), cadmium (Cd) is a suspected carcinogen, which may be present in air from industrial emissions and contaminate water. In the current study of cheese samples from the Latvian market, Cd was detected in only one sample of Roquefort cheese at the LOD level (0.002 mg kg^−1^ ww). Starska et al. (2011) also reported similar levels of Cd in dairy products, including cheeses from the Poland market [40]. Christophoridis et al. (2019) did not find traces of Cd or other toxic metals in blue cheese, whereas the concentrations in Feta cheese and Mozzarella marketed in Thessaloniki were comparable to that found in our study (1.15 and 2.95 µg kg^−1^ ww). Christophoridis et al. (2019) have provided a concise summary of recently published methods and test results of heavy metals in cheese samples. Thus, a wide range of Cd levels from 0.3 to 600 µg kg^−1^ ww have been reported in different cheese samples from Greece, Italy, Iran, and Egypt.

In the current study, lead (Pb) was found in more than 50% of the tested cheese samples (*n* = 25), with the concentrations ranging between 0.007 and 0.024 mg kg^−1^ ww. Most of the samples contained Pb at trace levels between the LOD and 0.017 mg kg^−1^, while only three samples (one DorBlu, Latvian blue cheese, and one Blue Stilton variety) contained concentrations around the maximum permissible level (0.02 mg kg^−1^ or 20 µg kg^−1^) established for milk according to the EU Regulation 1881/2006 [3]. Cristophoridis et al. (2019) found a considerably wider range of Pb levels (0.014–10.7 mg kg^−1^ ww) in cheese samples from Egypt, Brazil, Turkey, and other regions.

Tin (Sn) is another toxic metal, which may migrate from metal surfaces in machinery into milk, resulting in a rather high content (39.5 mg kg^−1^ ww) determined in Roquefort cheese, which was sold under a supermarket brand, whereas the other positive samples (*n* = 12) contained only traces of this element around the detection level. From the detected micronutrients, Zn was determined at levels up to 39.5 mg kg^−1^ ww, similar to those reported by other authors concerned with the bioavailability of micronutrients in the soil used for dairy farming and migration from machinery into milk [19]. The other micronutrients (Mn, Fe) were present in the samples at very low concentrations, which were comparable to those reported by [19].

### 3.5. Exposure Assessment and Risk Characterization

#### 3.5.1. Dietary Consumption Patterns

Acute consumption data are available for the age categories of “Infants” (Denmark), “Toddlers” (Denmark), “Adolescents” (Denmark, France), “Adults” (Denmark, Finland, France, Italy), and “Elderly” (Denmark, France). Chronic consumption data are available for the age categories of “Other children” (Denmark, France), “Adolescents” (Denmark, France), “Adults” (Denmark, Spain, France, Italy, United Kingdom), and “Elderly” (Denmark, France). Based on the acute consumption data, the most frequent consumers of blue-cheese were Danish adults and elderly (consuming on 9 to 14% of days), however, the P95 values of the consumed amounts were not high (0.20–0.24 g kg^−1^ bw day^−1^). The French consume blue cheese less frequently across the population groups (consuming on 1 to 5% of days), but the intake is much higher, reaching 1.45–1.68 g kg^−1^ bw day^−1^ of blue cheese in the case of adolescents and 1.2–1.68 g kg bw^−1^ day^−1^ in the case of adults and elderly. High consumption (P95) values were also observed for Danish infant and toddler population (1.05 and 0.79 g kg^−1^ bw day^−1^) and Italian adults (1.3 g kg^−1^ bw day^−1^). Finnish adults are occasional consumers (consuming on 3% of days), with the average consumption P95 value of 0.6 kg^−1^ bw day^−1^. The data are summarised in Appendix A.

#### 3.5.2. Toxicological Reference Doses

The reference dose (RfD) (mg kg^−1^ bw day^−1^) used for the HI calculations is the maximum daily dose of a compound from a specific exposure pathway that is believed not to lead to an appreciable risk of harmful effects to sensitive individuals. The oral chronic exposure RfD values for heavy metals (0.001, 0.0035, 0.005, 0.005, 0.14, 0.30, 0.60, 0.70, and 1.0 mg kg^−1^ bw day^−1^ for Cd, Pb, Se, Mo, Mn, Zn, Sn, Fe, and Al) were retrieved from EPA and WHO guidelines [41,42].

The threshold toxicological values of BAs were converted to acute oral reference doses (aRfD). To obtain the aRfD values, the toxicological doses (in mg) were adjusted to the weight of adults (to mg kg^−1^ bw day^−1^) by applying the average human body weight of 70 kg obtained from the EFSA database [43]. Since the weight adjustment for children and adolescents may lead to an underestimation of the toxicological dose (i.e., higher dose per body weight for the same toxic effect), the adult aRfDs were also applied to the subgroup analysis of children and adolescents. Most of the aRfDs for BAs were obtained from the EFSA Scientific Opinion on the risk-based control of biogenic amine formation in fermented foods [27]. A limited number of studies have reported the dose-response relationships of BAs; hence, the findings are rarely conclusive. In the case of His, EFSA has established a potential NOAEL (aRfD) level of 50 mg [27]. Tyr, Phe, and Try may cause similar toxic effects in humans. Generally, high amount (600–2000 mg) of Tyr must be administered to provide biological response in humans, but the individuals medicated with monoamine oxidase inhibitor (MAOI) drugs could be 20 to 56 times more sensitive to Tyr and other BAs. Based on the studies evaluated in the EFSA Scientific Opinion [27] and a recent review [19], we applied the aRfDs of 100, 2000, and 30 for Tyr, Try and Phe, respectively. No human dose-response data are available for Put and Cad, but a NOAEL of 180 mg kg^−1^ bw day^−1^ has been observed in a study on Wistar rats [27]. We converted the NOAEL by applying the interspecies safety factor of 10 to obtain an aRfD of 18 mg kg^−1^ bw day^−1^ for adults.

No prior toxicological studies have established a dose-response relationship of ROQ C and there are no literature reports on this subject, besides a study that confirms the interaction of ROQ C and mycophenolic acid, which acts as synergists in eliciting biological effects in human cells. A recent study described cytotoxicity tests of human monocytic and intestinal cells after exposure to ROQ C and MPA at 10 to 50% of inhibitory concentrations (i.e., IC10 and IC50), in order to establish potential toxic impact associated with acute dietary exposure [44]. According to that study, both mycotoxins safe to consumers as they did not induce apoptosis in Caco-2 cells after a 24 h period under the selected test conditions. In order to establish a possible toxicological threshold, we applied the TTC approach [45]. After analysis of ROQ C by Toxtree v3.1, it was classified as a Cramer Class III substance, for which the toxicological TTC value is 1.5 µg kg^−1^ bw day^−1^, and the same value was also assumed as aRfD according to EFSA recommendations [45].

MPA is mainly used therapeutically as an immunosuppressant drug. Generally, high therapeutic doses (1–3 g) are tolerated well in humans, with the major side effect being gastrointestinal intolerance [46]. Converting the minimum reported dose (1 g) to LOAEL of adult (70 kg) resulted in a value of 14 mg kg^−1^ bw day^−1^. Taking into account that NOAEL is one-tenth of the LOAEL, the aRfD was calculated for MPA as 1.4 mg kg^−1^ bw day^−1^.

For carcinogens, such as AFM_1_, a tolerable daily intake is generally not determined. It is recommended that the concentration of such compounds in food should be as low as reasonably achievable (ALARA) [47]. A reference dose of 570 ng kg^−1^ bw day^−1^ has been calculated for toxic effects in Fischer rats [47]. We converted the reference dose by applying an interspecies safety factor of 10 to obtain an aRfD of 57 ng kg^−1^ bw day^−1^ for humans.

There are insufficient data to establish a tolerable daily intake (TDI) and/or aRfD value for ENN B. According to the EFSA CONTAM panel, acute exposure to ENNB does not indicate concern for human health, but that may not be the case in the case of chronic exposure. Based on the repeated-dose ENNB study, the lowest NOAEL has been observed for female mice at 0.18 mg kg^−1^ bw day^−1^ for effects on thymus, uterus, and spleen [48]. We converted the NOAEL by applying an interspecies safety factor of 10 to obtain an aRfD of 18 µg kg^−1^ bw day^−1^ for humans.

#### 3.5.3. Exposure Assessment of Heavy Metals

Considering the potential long-term effects of heavy metals and other toxic elements, only chronic exposure was evaluated for heavy metals. Table 6 depicts the calculated hazard quotient values, which ranged between 0.01 to 0.14 for heavy metal exposure based on the available dietary data for different age group consumers from Denmark, France, Italy, Spain, and the United Kingdom (additional information provided in the Appendix A).

The lowest values were determined for Danish consumers, whereas the widest ranges between the average and high-level consumers were determined for Italian and Spanish adults. Because of the lack of consumption data, it was only possible to compare the different age groups of Denmark and France. Overall, the calculated hazard indices show that there is no risk for the general population across Europe based on the chronic consumption data, however the rather high HI values for the French population show that cheese might be one of the notable contributors to dietary heavy metal exposure.

#### 3.5.4. Exposure Assessment of Biogenic Amines

Considering the lack of proper knowledge about human health effects from acute dietary intake of BAs present in blue cheeses, a multi-step procedure on the basis of three different scenarios was employed. First, the hazard index values for chronic dietary exposure to BAs were evaluated for different groups of European consumers in Denmark, Finland, France, and Italy (Table 7).

While the first two scenarios showed similarly low hazard levels, as noted for chronic exposure impact of BAs (HI values ranged between 0.01 and 0.08), the high consumer values for France in some cases were notably higher compared to other countries (HI values above 0.08), while there were no notable differences between survey years. However, the worst-case scenario based on high consumption levels and P95 of occurrence data indicated 9 to 18 times higher HI values for all population groups in the evaluated countries, compared to the mean consumer data. As expected, the HI of acute exposure to BAs via the intake of blue cheese occurred within the worst-case scenario for younger age groups (i.e., infants > toddlers > adolescents), especially for the Danish population where the HI values were more than 4 times higher for infants compared to those for adults according to the survey data of years 2005–2006 (Appendix A).

In order to determine the impact of individual BAs on the possible risks following acute exposure, the data of BA intake were compared with the aRfD values for those consumer groups (Table 8 and the Appendix A).

The mean/high consumption and mean UB occurrence value scenarios indicated a low level of risk (the exposure to Tyr and His was merely 0.3 to 5% of the aRfD threshold). For other BAs, the risk was also low even in the case of worst-case scenario, due to the relatively low toxicity of Phe, Put, and Try in combination with the generally low concentration levels of these BAs in the analysed cheese samples.

However, the relatively higher exposure to His and Tyr according to the worst case acute consumption scenarios gave reasons for a serious concern regarding toxicological effects from these BAs. For both His and Tyr, the lowest values were determined for elderly and adult groups in Denmark, ranging between 3.2 and 4.5% of the aRfD. The same values were 4 to 5 times higher for Danish toddlers and infants (the values for His and Tyr in the case of Danish infants reached up to 16.7 and 25.2% of the aRfD threshold). Unfortunately, it was not possible to compare those data with the infant and toddler groups from other countries due to the lack of statistically significant consumption data. Among the elderly and adult groups, higher potential risk for toxicological effects due to His and Tyr intake were determined for the population of France and Italy, when compared to other countries: the worst-case scenarios amounted to 28.6–33.1% of aRfD for Tyr and 19.0–26.6% of RfD for His, respectively. The maximal exposure to His and Tyr reached 26.6% and 40.1% of the aRfD values in the adolescent group of French population in 2014. In contrast, the maximal exposure of this subgroup was 1.15 times lower in 2007, indicating a potential increase of BA related risks for consumer safety, which has been also noted in a study by [19] and associated with the increased amounts of BAs detected in blue mould cheeses in recently reported papers. While these values did not reach the aRfD, meaning that the determined BA levels pose no immediate danger to consumers, the performed evaluation indicates potential risk, especially for susceptible population groups, such as of HIV-positive individuals or patients taking antihistamine drugs.

#### 3.5.5. Exposure Assessment of Mycotoxins

The dietary exposure to M_1_, ENN B, and MPA poses no acute health risk to humans, based on the evaluation by percentage (%) of aRfD. The highest MPA exposure was observed in French adolescents at the level of 0.63 µg kg^−1^ bw day^−1^ (Appendix A), corresponding to 0.045% of aRfD. Similarly, the highest levels of exposure to AFM_1_ and ENN B were equivalent only to 0.08% and 0.013% of aRfD, respectively (data not shown). However, there might be some risk to consumers in the case of ROQ C. Our screening approach via TTC indicated that even for average consumers (Table 8 and Appendix A) the exposure to ROQ C was equal to a significant fraction of aRfD, based on our tentatively assigned aRfD value of 1.5 µg kg^−1^ bw day^−1^. For average consumers, the exposure ranged from 5% to 41% of the aRfD values in the case of elderly Danish and Italian adults. For high consumers the exposure was 2 to 5 times higher, ranging from 11% to 95% in the case of elderly Danish consumers and French adolescents. In the worst-case scenario, the exposures exceeded 2–3 times of the aRfD for 11 out of 16 population groups across different nations (Table 8). The highest exposures were observed for French adolescents (375% of aRfD) and adults (309% of aRfD).

Generally, high levels of ROQ C are accepted in cheese and there are many studies claiming that high levels of ROQ C pose no risk to human health via dietary consumption, however, proper dose-response relationship for toxicological effects has not yet been established. Clearly, there is a need for such studies.

## 4. Conclusions

The current food safety regulations regarding the European market impose rather limited controls of toxins and pollutants in dairy products of short shelf times, such as mould cheeses. Methods based on HPLC-MS/MS and PAD detection for the analysis of major mycotoxins in cheeses (ROQ C, MPA) and different spoilage indicators (fourteen additional mycotoxins, six biogenic amines) and ICP-MS for the analysis of heavy metals were validated and applied to forty-six blue cheese samples marketed in Latvia. The determined levels of mycotoxins characteristic for blue cheese (ROQ C, MPA) and the levels of histamine and other biogenic amines showed similar concentration ranges, as reported by some other recent studies. Meanwhile, three of the gorgonzola samples were also positive for Aflatoxin M1, concentrations up to the maximum allowed level for milk used in dairy products. One emerging mycotoxin (ENN B) was detected for the first time in blue mould cheeses at trace levels. Heavy metals were present in the tested cheese samples at very low concentrations. Deterministic modelling of chronic and acute dietary exposure to the determined mycotoxins, BAs, and heavy metals was performed on the basis of consumption data from different European countries and different age categories. Rather high acute hazard indexes were obtained for some BAs (HIs up to 0.77) according to the worst-case scenario based on high consumption and 95th percentile occurrence. More detailed acute dietary intake studies indicated that histamine and tyramine reached 27 and 41% of the acute oral intake reference doses, whereas other BAs were present at insignificant levels, even according to the worst-case scenarios, thus the analysed cheeses can be considered as safe for healthy adults. The tentative identification of risks due to ROQ C via TTC highlighted the need for additional studies to evaluate the dose-response relationships of ROQ C and related contaminants. Continuation of similar studies will provide new knowledge for food and environmental safety and may help future researchers and policy makers in evaluating safety issues associated with traditional food consumption.

## Figures and Tables

**Figure 1 foods-09-00093-f001:**
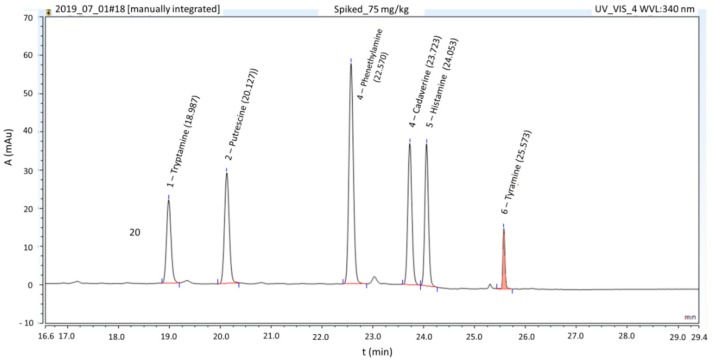
Representative chromatograms of biogenic amines in a blank sample spiked with standard mixture at 75 mg kg^−1^ concentration.

**Figure 2 foods-09-00093-f002:**
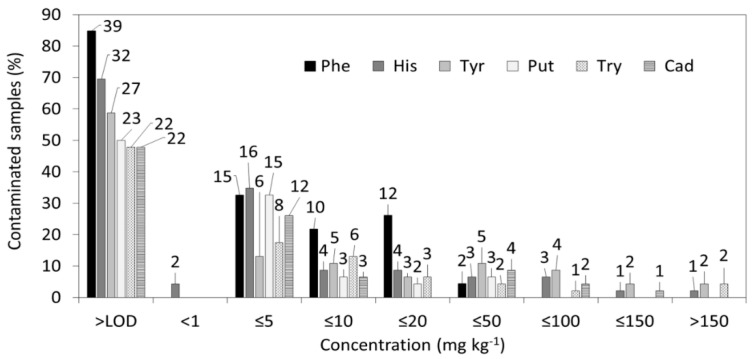
The relative distribution of blue mould cheese contamination with biogenic amines (count of positive samples for each biogenic amine is shown above each bar diagram.

**Table 1 foods-09-00093-t001:** Instrumental parameters of LC-MS/MS method.

Mycotoxin	RT (min)	Precursor (*m*/*z*)	MRM_1_ (*m*/*z*)	CE_1_ (*m*/*z*)	MRM_2_ (*m*/*z*)	CE_2_ (*m*/*z*)	MRM_3_ (*m*/*z*)	CE_3_ (*m*/*z*)
PA	1.0	171.1	125.0	13	153.0	10	−	−
AFM_1_	1.4	329.1	229.0	44	273.1	22	−	−
ROQ C	2.2	391.0	193.1	23	323.1	18	−	−
AFB_1_	2.5	313.0	240.9	39	285.0	22	−	−
MPA	5.0	319.0	191.0	18	205.0	16	−	−
OTB	5.4	370.1	103.1	30	205.0	20	−	−
CVD	6.1	420.4	285.2	14	315.1	10	−	−
SBL	6.5	386.3	122.1	55	150.0	55	178.0	40
OTA	6.5	404.0	239.0	22	358.0	15	−	−
CIT	7.2	251.1	191.1	16	205.1	15	−	−
PNA	7.9	634.4	558.5	20	616.4	12	−	−
ENN B	8.3	657.4	196.0	32	214.0	34	−	−
ENN B1	8.5	671.3	196.0	34	214.0	35	−	−
BEA	8.5	802.7	244.1	25	785.3	20	−	−
ENN A	8.6	685.2	210.0	31	228.0	33	−	−
ENN A_1_	8.8	699.4	210.0	34	228.0	35	−	−

RT—retention time; MRM_1_: quantifier transition; MRM_2_, MRM_3_: qualifier transitions; CE: collision energy.

**Table 2 foods-09-00093-t002:** Instrumental parameters of ICP-MS method.

Parameter	Value
Plasma mode	normal, robust
RF power (kW)	1.30
Sampling depth (mm)	8.0
Carrier gas flow (L min^−1^)	0.6
Dilution gas flow (L min^−1^)	0.4
Spray chamber temperature (°C)	2.0
Extraction lens l (V)	0
Kinetic energy discrimination (V)	3
Scanning mode	Peak hopping

**Table 3 foods-09-00093-t003:** The concentrations of mycotoxins found in blue cheese samples.

Country	ROQ-C (µg kg^−1^)			MPA (µg kg^−1^)			AFM_1_ (µg kg^−1^)		
	Cases	Mean	Range	Cases	Mean	Range	Cases	Mean	Range
Italy	11/11	836	144–2542	6/11	10.9	9.4–15.7	3/11	0.03	0.02–0.05
France	9/9	396	3.2–718	5/9	156	9.5–208	−	−	−
Poland	7/7	439	5.3–790	2/7	7.7	5.8–9.5	−	−	−
Denmark	7/7	1051	209–2150	7/7	208	9.7–599	−	−	−
England	3/3	3675	2077–5454	1/3	−	9.6	−	−	−
Latvia	3/3	620	101–1323	3/3	23.9	10.7–41.3	−	−	−
Spain	2/2	138	111–165	2/2	108	91–125	−	−	−
Lithuania	2/2	352	156–547	1/2	−	179	−	−	−
Germany	2/2	992	507–1476	2/2	20.8	9.7–31.8	−	−	−

**Table 4 foods-09-00093-t004:** The concentrations of biogenic amines found in blue cheese samples.

Biogenic Amine (BA)	Cases (Total %)	Uncontaminated Samples (*n*)	Mean Concentration (mg kg^−1^)	Concentration Range (mg kg^−1^)
Tyramine	58.7	27	70.7	1.1–717
Tryptamine	47.8	22	30.4	1.2–213
Putrescine	52.2	24	9.3	1.3–45.5
Phenethylamine	84.8	39	10	1.5–43.9
Cadaverine	47.8	22	22.1	1.7–131
Histamine	69.6	32	23.3	0.2–186

**Table 5 foods-09-00093-t005:** The concentrations of heavy metals found in blue cheese samples.

Heavy Metal	Cases (Total %)	Samples (n)	Mean Concentration (mg kg^−1^)	Concentration Range (mg kg^−1^)
Al	40.9	18/44	0.55	0.33–1.11
Mn	86	38/44	0.16	0.083–0.63
Fe	52.2	14/44	2.68	1.57–12.4
Zn	100	44/44	27.3	10.4–39.5
Se	100	44/44	0.11	0.035–0.29
Cd	2.3	1/44	0.002	0.002
Sn	29.5	13/44	2.59	0.024–32.5
Pb	56.8	25/44	0.013	0.007–0.024

**Table 6 foods-09-00093-t006:** Hazard indices for chronic dietary exposure to heavy metals from blue cheese.

**Country, Population and Survey Year**	**Denmark** **(Other Children, 2000)**	**Denmark (Adolescents, 2000)**	**Denmark (Adults, 2000)**	**Denmark (Elderly, 2000)**	**Spain (Adults, 2009)**	**France (Adults, 2007)**	**France (Adults, 2014)**	**France (Other Children, 2014)**	**France (Adolescents, 2007)**
HI average consumers	0.01	0.01	0.01	0.00	0.04	0.03	0.04	0.05	0.03
HI high consumers	0.02	0.01	0.02	0.01	0.11	0.07	0.09	0.14	0.06
**Country, Population and Survey Year**	**France (Adolescents, 2014)**	**France (Adults, 2007)**	**France (Elderly, 2007)**	**France (Adults, 2007)**	**France (Adults, 2014)**	**France (Adults, 2014)**	**France (Elderly, 2014)**	**Italy (Adults, 2005)**	**United Kingdom (Adults, 2000)**
HI average consumers	0.05	0.03	0.03	0.03	0.04	0.04	0.04	0.05	0.01
HI high consumers	0.10	0.07	0.08	0.07	0.10	0.08	0.09	0.09	0.03

**Table 7 foods-09-00093-t007:** Hazard indices for chronic dietary exposure to biogenic amines from blue cheese.

**Country, Population and Survey Year**	**Denmark (Infants, 2006)**	**Denmark (Toddlers, 2006)**	**Denmark (Adolescents, 2000)**	**Denmark (Adults, 2000)**	**Denmark (Adults, 2005)**	**Denmark (Elderly, 2000)**	**Denmark (Elderly, 2005)**	**Finland (Adults, 2007)**
HI average consumers	0.03	0.02	0.01	0.01	0.01	0.01	0.01	0.02
HI high consumers	0.08	0.06	0.02	0.02	0.02	0.01	0.02	0.05
HI high consumers, 95th percentile occurrence data	0.48	0.36	0.13	0.11	0.12	0.09	0.11	0.28
**Country, Population and Survey Year**	**Finland (Adults, 2012)**	**France (Adolescents, 2007)**	**France (Adolescents, 2014)**	**France (Adults, 2007)**	**France (Adults, 2007)**	**France (Adults, 2014)**	**France (Elderly, 2007)**	**Italy (Adults, 2005)**
HI average consumers	0.02	0.05	0.05	0.05	0.05	0.04	0.05	0.05
HI high consumers	0.04	0.11	0.12	0.10	0.10	0.09	0.10	0.10
HI high consumers, 95th percentile occurrence data	0.27	0.66	0.77	0.63	0.64	0.55	0.59	0.59

**Table 8 foods-09-00093-t008:** Acute dietary exposure to biogenic amines and Roquefortine C (RoQ C) from blue cheese, as percentage of aRfD values.

Country, Population and Survey Year	Tyr	Try	Put	Phe	Cad	His	ROQ C	Tyr	Try	Put	Phe	Cad	His	ROQ C	Tyr	Try	Put	Phe	Cad	His	ROQ C
	Mean Consumption, Mean UB Occurrence Values							High Consumption, Mean UB Occurrence Values							High Consumption, 95th Percentile Occurrence Values						
DK (Infants, 2006)	1.2%	0.0%	0.0%	0.8%	0.0%	0.9%	23%	3.2%	0.1%	0.0%	2.1%	0.1%	2.4%	60%	25.2%	0.6%	0.2%	5.2%	0.4%	16.7%	235%
DK (Toddlers, 2006)	1.0%	0.0%	0.0%	0.6%	0.0%	0.7%	18%	2.4%	0.0%	0.0%	1.6%	0.0%	1.8%	44%	18.8%	0.4%	0.2%	3.9%	0.3%	12.4%	175%
DK (Adolescents, 2000)	0.4%	0.0%	0.0%	0.3%	0.0%	0.3%	8%	0.9%	0.0%	0.0%	0.6%	0.0%	0.7%	16%	6.8%	0.2%	0.1%	1.4%	0.1%	4.5%	64%
DK (Adults, 2000)	0.3%	0.0%	0.0%	0.2%	0.0%	0.2%	6%	0.7%	0.0%	0.0%	0.5%	0.0%	0.5%	14%	5.7%	0.1%	0.1%	1.2%	0.1%	3.8%	54%
DK (Adults, 2005)	0.4%	0.0%	0.0%	0.2%	0.0%	0.3%	7%	0.8%	0.0%	0.0%	0.5%	0.0%	0.6%	15%	6.2%	0.1%	0.1%	1.3%	0.1%	4.1%	57%
DK (Elderly, 2000)	0.3%	0.0%	0.0%	0.2%	0.0%	0.2%	5%	0.6%	0.0%	0.0%	0.4%	0.0%	0.5%	11%	4.8%	0.1%	0.0%	1.0%	0.1%	3.2%	45%
DK (Elderly, 2005)	0.3%	0.0%	0.0%	0.2%	0.0%	0.3%	6%	0.7%	0.0%	0.0%	0.5%	0.0%	0.5%	13%	5.5%	0.1%	0.0%	1.1%	0.1%	3.7%	52%
FI (Adults, 2007)	0.9%	0.0%	0.0%	0.6%	0.0%	0.7%	16%	1.8%	0.0%	0.0%	1.2%	0.0%	1.4%	35%	14.6%	0.3%	0.1%	3.0%	0.3%	9.7%	136%
FI (Adults, 2012)	0.9%	0.0%	0.0%	0.6%	0.0%	0.7%	18%	1.8%	0.0%	0.0%	1.2%	0.0%	1.4%	34%	14.2%	0.3%	0.1%	2.9%	0.3%	9.4%	133%
FR (Adolescents, 2007)	2.1%	0.0%	0.0%	1.4%	0.0%	1.6%	40%	4.4%	0.1%	0.0%	2.9%	0.1%	3.3%	82%	34.7%	0.8%	0.3%	7.1%	0.6%	23.0%	324%
FR (Adolescents, 2014)	2.1%	0.0%	0.0%	1.4%	0.0%	1.6%	40%	5.0%	0.1%	0.0%	3.3%	0.1%	3.8%	95%	40.1%	0.9%	0.4%	8.3%	0.7%	26.6%	375%
FR (Adults, 2007)	2.0%	0.0%	0.0%	1.3%	0.0%	1.5%	38%	4.1%	0.1%	0.0%	2.8%	0.1%	3.2%	78%	33.0%	0.7%	0.3%	6.8%	0.6%	21.9%	308%
FR (Adults, 2007)	1.9%	0.0%	0.0%	1.3%	0.0%	1.4%	36%	4.2%	0.1%	0.0%	2.8%	0.1%	3.2%	78%	33.1%	0.7%	0.3%	6.8%	0.6%	22.0%	309%
FR (Adults, 2014)	1.6%	0.0%	0.0%	1.1%	0.0%	1.2%	30%	3.6%	0.1%	0.0%	2.4%	0.1%	2.7%	68%	28.6%	0.6%	0.3%	5.9%	0.5%	19.0%	268%
FR (Elderly, 2007)	1.9%	0.0%	0.0%	1.3%	0.0%	1.4%	36%	3.8%	0.1%	0.0%	2.6%	0.1%	2.9%	73%	30.7%	0.7%	0.3%	6.3%	0.5%	20.3%	286%
IT (Adults, 2005)	2.2%	0.0%	0.0%	1.5%	0.0%	1.7%	41%	3.9%	0.1%	0.0%	2.6%	0.1%	3.0%	74%	31.0%	0.7%	0.3%	6.4%	0.5%	20.6%	290%

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
