# Peer review of "The Occurrence and Dietary Exposure Assessment of Mycotoxins, Biogenic Amines, and Heavy Metals in Mould-Ripened Blue Cheeses"

_foods, 2020, doi:10.3390/foods9010093_

Round 1

Reviewer 1 Report

The findings in the study have provided important information on levels of mycotoxins, biogenic amines and heavy metals for health risk assessment from exposure of cheese consumption in a population.

Analysis and method validation:

The spike recoveries were only conducted for blanks, for both mycotoxins and biogenic amines. It would be useful if the actual cheese samples are spiked with these compounds and this would match for any matrix effect during extraction and analysis.

Similarly, the spike recoveries for heavy metal analysis were carried out only for blanks and not for actual cheese samples during microwave digestion and analysis. For heavy metal analysis in foods, there are many certified reference materials (CRMs) that can be obtained from suppliers (e.g. NIST, ERM, IAEA) for use as quality control during method validation.

Have any of the relevant CRMs been used at some stage for method validation and development?

Conclusions

In lines 752 to 754, the sentence may need to be clarified and re-written as "Continuation of similar studies will provide new knowledge for food and environmental safety and may help future researchers and policy makers in evaluating safety issues associated with traditional food consumption".

Author Response

The authors of the manuscript appreciate all the constructive suggestions made by the reviewer.

Analysis and method validation:

- In fact, several cheese varieties were evaluated for suitability as blank matrices with low initial probability of mycotoxin and biogenic amine contamination while having similar physicochemical characteristics to blue mold cheese varieties. Brine-matured fresh salad cheese “Grikios” with 45% fat content and no detectable levels of mycotoxins and biogenic amines was selected as the most suitable matrix, which was used in the instrumental method development and quality control analysis of mycotoxins and biogenic amines (Clarification added in lines 95-97, 150-151, 211-213)

- For heavy metal analysis, a standard addition method was used: daily analyses of quality control materials (with the concentrations of 2 μg/L and 50 μg/L) were performed for monitoring the repeatability and accuracy (clarified in lines 267-268). According to our knowledge, there is no CRM material available for heavy metal determination in mold cheeses. The calculation of the individual expanded method uncertainties for the individual chemical elements was provided using data of interlaboratory test results, which included various control reference materials (clarified in lines 269-271). The reviewer's recommendations for the use of appropriate reference materials based on cheese matrix will be taken into account in further research.

In Conclusion part the sentence in lines 753-755 was corrected according to the suggestion made by the the reviewer.

Reviewer 2 Report

The manuscript entitled ‘The occurrence and dietary exposure assessment of mycotoxins, biogenic amines, and heavy metals in mould-ripened blue cheeses’ investigates the occurrence and dietary exposure assessment of 16 mycotoxins, 6 biogenic amines, and 13 metallic elements in blue-veined cheeses. The subject undertaken in the study if of high importance for protection of public health. The methodology of the study was carefully design and detailed survey was performed. The results are presented clearly supported by unambiguous conclusions.

To conclude, I recommend the manuscript entitled ‘The occurrence and dietary exposure assessment of mycotoxins, biogenic amines, and heavy metals in mould-ripened blue cheeses.

Author Response

The authors are grateful to the reviewer for the positive evaluation of this article.